# The GluN2B-Selective Antagonist Ro 25-6981 Is Effective against PTZ-Induced Seizures and Safe for Further Development in Infantile Rats

**DOI:** 10.3390/pharmaceutics13091482

**Published:** 2021-09-16

**Authors:** Pavel Mareš, Lucie Kozlová, Anna Mikulecká, Hana Kubová

**Affiliations:** 1Department of Developmental Epileptology, Institute of Physiology, Czech Academy of Sciences, 14220 Prague, Czech Republic; Pavel.Mares@fgu.cas.cz (P.M.); luciee.kozlova@gmail.com (L.K.); anna.mikulecka@fgu.cas.cz (A.M.); 2Department of Rehabilitation and Sport Medicine, 2nd Medical Faculty, Charles University, 15006 Prague, Czech Republic

**Keywords:** development, GluN2B-selective antagonist, Ro 25-6981, anti-seizure effects, motor performance, memory, immature rats

## Abstract

The GluN2B subunit of NMDA receptors represents a perspective therapeutic target in various CNS pathologies, including epilepsy. Because of its predominant expression in the immature brain, selective GluN2B antagonists are expected to be more effective early in postnatal development. The aim of this study was to identify age-dependent differences in the anticonvulsant activity of the GluN2B-selective antagonist Ro 25-6981 and assess the safety of this drug for the developing brain. Anticonvulsant activity of Ro 25-6981 (1, 3, and 10 mg/kg) was tested in a pentylenetetrazol (PTZ) model in infantile (12-day-old, P12) and juvenile (25-day-old, P25) rats. Ro 25-6981 (1 or 3 mg/kg/day) was administered from P7 till P11 to assess safety for the developing brain. Animals were then tested repeatedly in a battery of behavioral tests focusing on sensorimotor development, cognition, and emotionality till adulthood. Effects of early exposure to Ro 25-6981 on later seizure susceptibility were tested in the PTZ model. Ro 25-6981 was effective against PTZ-induced seizures in infantile rats, specifically suppressing the tonic phase of the generalized tonic–clonic seizures, but it failed in juveniles. Neither sensorimotor development nor cognitive abilities and emotionality were affected by early-life exposure to Ro 25-6981. Treatment cessation did not affect later seizure susceptibility. Our data are in line with the maturational gradient of the GluN2B-subunit of NMDA receptors and demonstrate developmental differences in the anti-seizure activity of the GluN2B-selective antagonist and its safety for the developing brain.

## 1. Introduction

Epilepsy is the most common neurological disorder in children. Epidemiological studies show that seizures and epilepsy affect infants and neonates more than any other age group. Early life epilepsies are frequently associated with a broad spectrum of cognitive, behavioral, and mental health comorbidities. Likely, a common pathophysiological substrate is responsible for both epilepsy and comorbidities and targeting of the underlying mechanisms can effectively treat both these conditions [1].

Pharmacological intervention with antiepileptic drugs (AEDs) is the mainstay of treatment of childhood epilepsies. However, it is important to recognize that many classical AEDs have serious adverse effects that can exacerbate epilepsy comorbidities in pediatric patients. In addition, when administered to pregnant women, some of them can affect normal brain development and result in temporary and permanent functional alterations [2]. As suggested by experimental studies, anti-seizure as well as the adverse effects of AEDs can be dependent on the developmental stage of the brain and vary with age. Thus, the safety and efficacy of AEDs is a major concern of clinicians taking care of children with epilepsy. Despite this, the only limited attempt has been made to develop age-specific AEDs, and possible adverse effects of clinically used AEDs on brain development are only rarely assessed [3,4].

Recent literature documents that the immature brain differs considerably from the adult brain in the generation, propagation, and termination of seizures, in their EEG, and in their behavioral features and consequences (for reviews, see [5,6]), as well as in the mechanisms underlying the development of epilepsy. The neonatal period is the time of high risk for seizures, and numerous experimental studies have documented an increased seizure propensity. Animal experiments have demonstrated increased excitability of neurons and neuronal networks in the neonatal and infantile brain. On one side, increased excitability substantially contributes to activity-dependent synaptogenesis and brain wiring, on the other side, it predisposes the immature brain to seizures. The enhanced excitability of the immature brain is related to the sequential development of the essential neurotransmitter signaling pathways—glutamatergic and GABAergic. Glutamate is the principal excitatory transmitter in the brain, and it excites target neurons via ionotropic receptors (N-methyl-D-aspartate (NMDA), a-amino-3-hydroxy-5-methylisoxazole-4-proprionic acid (AMPA), and kainite receptors). At the early stages of postnatal development, glutamatergic excitation is enhanced due to an overabundance of NMDA receptors. Due to their high Ca^2+^ permeability, voltage dependence, and slow kinetics, these receptors play a crucial role in neuroplasticity and the development of neuronal networks. Whereas excessive activation of NMDA receptors at early stages of development results in seizures and neuronal loss, their antagonists exhibit a potent anticonvulsant action. Unfortunately, many previously tested NMDA antagonists also produce undesirable side effects at doses within their therapeutic range. Both acute adverse effects on brain functions and long-lasting adverse effects on brain development have been reported.

NMDA receptors are heterotetramers with two obligatory type 1 subunits (GluN1) and two other subunits. The most frequent composition is made from two GluN1 and two GluN2 subunits that contribute substantially to the functional diversity of the NMDA receptors. The GluN2 subunits are classified as the A–D types, with GluN2A and GluN2B being more common than the other two. Individual subtypes of the GluN2 subunits are distinctly expressed during development and among the brain regions. The GluN2B subunit predominates at the early stages of postnatal development in rats. The representation of the GluN2A subunit gradually increases during the first three postnatal weeks, becoming the dominant subunit in mature receptors [7,8,9]. Thus, subunit-selective antagonists and modulators targeting the different GluN2 subunits might provide an opportunity to find age-specific antiepileptic drugs. As documented previously, ifenprodil, a noncompetitive antagonist, working on the GluN2B subunit [10,11], is effective in various seizure models in adults [12,13,14,15] as well as in developing rats [16,17]. A study on the adverse effects of ifenprodil has shown mild motor impairment within 30 min after administration in both infantile and juvenile rats [16].

In our previous study, PEAQX, a GluN2A-preferring antagonist of NMDA receptors, exhibited anti-seizure activity in infantile rats [18]. Selectivity of PEAQX to GluN1/GluN2A receptors vs. GluN1/GluN2B receptors is relatively low and PEAQX in high doses effectively suppressed seizures also in juvenile P25 animals. Ro 25-6981 [4-((1R,2S)-3-(4-benzylpiperidin-1-yl)-1-hydroxy-2-methylpropyl)phenol] is a highly potent and selective blocker of the GluN2B subunit containing NMDA receptors [19,20]. The in vivo efficacy of Ro 25-6981 was confirmed in many studies in which neuroprotective [21], antidepressant [22], and anti-nociceptive [23] effects have been demonstrated. Knowledge concerning the anti-seizure effects of Ro 25-6981 and their age-dependent changes are, however, limited.

The aim of the present study was to determine the anticonvulsant potential of the GluN2B blockade and its age-dependence in developing rats together with its safety for developing brain. In the first part of the present study, we aimed to assess the acute anticonvulsant effects of Ro 25-6981 in two age groups, 12-day-old (P12), and 25-day-old (P25) rats. The possible adverse effects of a single dose of Ro 25-6981 on memory were tested in P25 rats. The second part of this study was designed to determine the possible deteriorating effects of early exposure to Ro 25-6981 on seizure susceptibility, development of sensorimotor functions, motor coordination, emotional behavior, and cognitive abilities. In this part of the study, Ro 25-6981 was administered for five consecutive days between P7 and P11, and animals were repeatedly tested in a battery of behavioral tests from postnatal day 12 to early adulthood (P60). Additional groups of animals were used to determine the effects of early exposure to Ro 25-6981 on memory in juvenile animals (P25) and on seizure susceptibility of infantile and juvenile rats (P12 and P25).

## 2. Materials and Methods

### 2.1. Animals

Experiments were performed on male Wistar albino rat pups (*n* = 261, Institute of Physiology of the Czech Academy of Sciences). The day of birth was defined as day 0, and animals were weaned at postnatal day (P) 21. Rats were housed in a controlled environment (temperature 22 ± 1 °C, humidity 50–60%, lights on 6 am–6 pm) with free access to food and water. During experiments with P12 pups, the temperature in Plexiglas cages was maintained at 32 ± 2 °C using an electric heating pad connected to a digital thermometer, placed on the bottom of cages, to compensate for the immature thermoregulatory functioning at this age. Before experiments, animals were transferred from animal room to the special rooms located in the same corridor and adapted to the novel environment for at least 30 min.

All procedures involving animals and their care were conducted according to the ARRIVE guidelines https://www.nc3rs.org.uk/the-3rs/ (accessed on 9 April 2021) in compliance with national (Act No 246/1992 Coll.) and international laws and policies (EU Directive 2010/63/EU for animal experiments and the National Institutes of Health guide for the care and use of Laboratory animals NIH Publications No. 8023, revised 1978). The experimental protocol was approved by the Animal Care and Use Committee of the Czech Academy of Sciences (Approval No. 15/2018).

### 2.2. Drugs

Ro 25-6981 maleate ((αR, βS)-α-(4-hydroxyphenyl)-β-methyl-4-(phenylmethyl)-1-piperidinepropanol maleate) was purchased from Tocris Biosciences (# 1594/1, Bristol, England) and dissolved in physiological saline in the concentration of 3 mg/mL and injected intraperitoneally. Controls received a corresponding volume of solvents (i.e., saline) instead of Ro 25-6981 solution. Pentylenetetrazol (PTZ, Sigma # P6500, St. Louis, MO, USA) was dissolved in distilled water in the concentration of 50 or 30 mg/mL to inject always a volume of 2 mL/kg.

Solutions were always freshly prepared before the injection.

### 2.3. Experimental Design

The present study consists of two parts. The first part aimed to assess the anticonvulsant as well as adverse effects of Ro 25-6981 administered acutely in a single dose to infantile and juvenile animals. The second part focused on the effects of early life sub-chronic exposure to Ro 25-6981 on further development, seizure susceptibility, and brain functions later in life.

#### 2.3.1. Acute Effects of Ro 25-6981

##### Anticonvulsant Effects

Acute anticonvulsant effects of Ro 25-6981 in three doses 1, 3, and 10 mg/kg were assessed using a PTZ model in infantile (P12) and juvenile (P25) rats. PTZ in a dose of 100 mg/kg was injected subcutaneously 30 min after Ro 25-6981 administration. Controls received physiological saline. At this dose, PTZ elicits two types of convulsive seizures, which differ by developmental profile, seizure generator, and pharmacological sensitivity. The appearance of minimal (mS, mostly clonic convulsions involving head and forelimbs muscles with preserved righting reflexes) seizures is age-dependent. In the PTZ model, minimal seizures occur in animals older than 15 days, but not in younger rats [24], whereas generalized tonic–clonic seizures (GTCS), starting with a short running phase and accompanied by a loss of righting reflexes, can be induced in all age groups of rodents starting at P1 [25].

After PTZ injection, animals were individually placed in Plexiglas cages and observed by an experienced observer for 30 min. Incidence and latency of both seizure types (mS and GTCS) and other behavioral phenomena (e.g., isolated myoclonic jerks) and abnormalities were also recorded. To assess the severity of the epileptic phenomena, animals were assigned a score for the most severe behavioral characteristics as follows [18].

0—no changes;0.5—abnormal behavior (e.g., automatisms, increased orienting reaction);1—isolated myoclonic jerks;2—typical minimal seizures or some parts of their symptomatology;3—clonic seizures (mS) involving head and forelimb muscles with preserved righting reflexes (older term minimal metrazol seizures);4—generalized seizures without the tonic phase (GCS);5—complete generalized tonic-clonic seizures (GTCS).

Individual dose groups consisted of 8 animals except for the 10 mg/kg group (*n* = 10 in P12 and *n* = 9 in the P25 group). The animals used for this experiment originated from at least eight litters.

##### Adverse Side Effects on Passive-Avoidance Responding

P25 rats received Ro 25-6981 in doses of 3 and/or 10 mg/kg intraperitoneally (*n* = 9), 30 min before the 1st exposure. Control animals (*n* = 8) received saline. The animals were subjected to a three-trial, step-through, passive-avoidance paradigm. The apparatus (Ugo Basile, Gemonio, Italy) was a rectangular Plexiglas cage (52 cm × 30 cm × 35 cm) consisting of two boxes separated by a sliding door, one being lit, while the other dark. At the training (1st trial), the rat was placed in the light box for 60 s, then the door was opened, and the rat escaped to the dark box. The door was shut closed, and a 2.0 mA foot shock was delivered for 5 s. Retention of the experience (without shock) was determined by repeating the test and recording the latency to step through after 1 and 24 h (2nd and 3rd trial, respectively).

#### 2.3.2. Effects of Early-Life Exposure to Ro 25-6981 on Brain Functions and Seizure Susceptibility

This experiment aimed to detect possible adverse effects of early brain exposure to Ro 25-6981 persisting after treatment cessation. Ro 25-6981 was administered in doses 1 or 3 mg/kg/day starting at P7 till P11 (i.e., for five consecutive days). Body weight was checked daily from P6 to P13 to assess the possible effects of repeated administration of Ro 25-6981 on weight gain. These data were used to calculate relative body weight (body weight at P6 was taken as 100%) to minimize variability in individual groups. The difference in relative body weights between two consecutive days was used as a measure of weight gain.

Twelve litters of animals were used for this experiment. Pups in each litter (*n* = 10) were randomly divided into three treatment groups, and all treatments were represented proportionally (3:3:4) to have 8 to 10 rats in each age and dose group.

##### Long-Term Effects on Seizure Susceptibility

Possible changes in seizure susceptibility were assessed using a model of PTZ-induced seizures (Figure 1). PTZ was injected at P12 and P25 in the dose of 60 mg/kg and/or 100 mg/kg. The following groups of animals were constituted: control (*n* = 8); 1 mg/kg (*n* = 8); 3 mg/kg (*n* = 10), respectively. For the PTZ at the dose of 100 mg/kg, the following groups of animals were constituted: control (*n* = 10); 1 mg/kg (*n* = 8), and/or 3 mg/kg of Ro 25-6981 (*n* = 8). Tests were performed as described above.

##### Long-Term Effects on Passive Avoidance Responding

Possible long-term effects on passive avoidance responding were assessed in P25 animals in the procedure described above (controls (*n* = 10); 1 mg/kg (*n* = 10); and 3 mg/kg (*n* = 9)) (Figure 2). This test was performed in a separate group of animals.

##### Long-Term Effects on Sensorimotor Performance and the Open-Field Behavior

In total, 30 animals (controls (*n* = 10), 1 mg/kg (*n* = 10), and 3 mg/kg (*n* = 10)) from three litters were used for this experiment. The animals were tested repeatedly at the age of 12, 15, 18, 21, 25, 31, and 60 days. The tests were chosen according to the sensorimotor development at individual ages (for details, see Figure 3) and in the open-field test. The sequence of tests was always the same. All animals were tested first in the open field. The animal was placed into the center of the field (40 cm × 40 cm × 15 cm) and distance traveled, the velocity of walking, and the frequency of the center visit and time spent in the center of the field as an index of anxiety-like behavior were evaluated with EthoVision software (Noldus Information Technology, Wageningen, The Netherlands). The following sensorimotor tests were performed: surface righting reflex, negative geotaxis, and bar holding in the 12- to 21-day-old groups, rotarod in 25-day-old rats, and the ladder rang walking test from 21-day-old rats (Figure 1). The time to cross the entire length of the ladder (100 cm) was assessed in two trials, the first with regular gaps and the second with irregular gaps. For the regular arrangements, the rungs were spaced at 2 cm intervals. For the irregular pattern, the distance of the rungs varied from 1 to 5 cm. In addition, the mean number of foot slips in foot placement was calculated. A detailed description of the individual tests was published previously [26,27].

### 2.4. Statistics

The sample size was determined in advance according to previous experience with the given tests and followed the principles of the three R’s (Replacement, Reduction, and Refinement; https://www.nc3rs.org.uk/the-3rs, accessed on 9 April 2021). Outcome measures and statistical tests were prospectively selected. At the beginning of the study, simple randomization was used to assign each animal to a particular treatment group. Data acquisition and analysis were done blinded to the treatment. Data were analyzed using GraphPad Prism 8 (GraphPad Software, San Diego, CA, USA) software. Using the D’Agostino–Pearson normality test, all the data sets were first analyzed to determine whether the values were derived from a Gaussian distribution. Data sets that did not meet strict normality criteria were analyzed using a Kruskal–Wallis test. One-way ANOVA and two-way repeated-measures ANOVA were used to identify the main effect of the drugs. In the presence of a significant main effect but without interaction between factors, simple effects were considered. Whenever a significant interaction was identified, the data were subjected to Tukey’s post-hoc test for multiple comparisons and the False Discovery Rate (FDR) was calculated to correct the p-value. Seizure severity or latency between two groups of controls receiving either 60 or 100 mg/kg of PTZ were compared with either a Mann–Whitney or an unpaired t-test. Incidence of individual seizure phenomena was compared first with χ^2^-test for trends and subsequently the control and individual dose groups using a Fisher exact test. Mixed effect analysis was used instead of a two-way repeated measure ANOVA if there were missing values; *p*-value < 0.05 was required for significance and *q* < 0.05 was taken as discovery.

## 3. Results

### 3.1. Acute Effects of Ro 25-6981

#### 3.1.1. Anticonvulsant Effects

Subcutaneous administration of PTZ in a dose of 100 mg/kg reliably evoked generalized tonic–clonic seizures in the P12 controls. All vehicle-pretreated rats developed generalized tonic–clonic seizures (GTCS). In this age group, administration of PTZ does not induce minimal, predominantly clonic seizures with preserved righting reflexes [24]. Ro 25-6981 administered 30 min before PTZ exhibited prominent anticonvulsant effects in all three tested doses (Figure 4). In the controls, PTZ produced an average seizure score of 5, and Ro 25-6981 in all three tested doses significantly decreased seizure severity, as revealed with Kruskal–Wallis test (*H* = 18.98, *p* < 0.0001) (Figure 4A). Predominantly, Ro 25-6981 protected against the tonic phase of GTCS and in doses 3 and 10 mg/kg decreased its incidence as revealed with a Fisher’s test (*p* = 0.007 and *p* < 0.0001) (Figure 4C). In addition, Ro 25-6981 prolonged the latency to GTCS (*F*_(3, 32)_ = 8.00, *p* = 0.004) and this effect was driven by increased latency in 3 and 10 mg/kg groups (Figure 4B).

In vehicle-treated P25 controls, PTZ in a dose of 100 mg/kg induced both generalized and minimal seizures in all animals in the group (Figure 5). Complete GTSC were observed in 8 of 8 control animals. Neither seizure severity (*H* = 3.554, *p* = 0.3138, Kruskal–Wallis) nor latency to GTCS (*F* _(4, 31)_ = 1.718, *p* = 0.6329, ANOVA) differ across groups (Figure 5). There was a near-significant effect of drug treatment on the incidence of the tonic phase of GTCS (χ^2^ test for trends (3.561,1, *p* = 0.0592), Figure 5C1) driven by the effect of the highest dose of 10 mg/kg. Pretreatment with Ro 25-6981 had no significant effect on minimal, predominantly clonic seizures with preserved righting reflexes (Figure 5).

#### 3.1.2. Adverse Side Effects on Passive Avoidance Responding

The effects of Ro 25-6981 alone on performance in the passive avoidance test were studied only in juvenile (P25) animals. As revealed with two-way ANOVA performance of animals in this test was not affected by acute pretreatment with Ro 25-6981 at any interval (effect of treatment *F*_(2, 24)_ = 0.65, *p* = 0.52). This result indicates that the drug did not affect fear-related memories (effect of interval *F*_(2, 24)_ = 79.56, *p* < 0.0001). (Figure 6).

### 3.2. Effects of Early Life Exposure to Ro 25-6981 on Brain Functions and Excitability

Effect of daily administration (P7–P11) of Ro 25-6981 on body weight was determined between P7 and P13. Absolute body weight increased, whereas relative weight gain decreased with age in all treatment groups (effect of age *F*_(7, 47)_ = 3062, *p* < 0.0001 and *F*_(3, 47)_ = 7.539, *p* = 0.0008, respectively). Neither absolute body weight nor relative weight gain between two consecutive days was affected by repeated drug exposure, as revealed by the mixed effect analysis (Figure 7).

#### 3.2.1. Long-Term Effects on Seizure Susceptibility

We next determined whether repeated administration of Ro 25-6981 at P7–P11 affects seizure susceptibility at two intervals after treatment cessation (Figure 1). At P12 or P25 rats were subcutaneously injected with PTZ in two different doses (60 mg/kg or 100 mg/kg). In vehicle-treated animals, both doses of PTZ-induced seizures of the same severity at P12 (*U* = 33, *p* = 0.6078) as well as at in P25 (*U* = 29.5, *p* = 0.1192). In P12 rats, latency to generalized seizures was shorter numerically but not statistically (*t* = 2.031, *df* = 16, *p* = 0.0592). In addition, latency to minimal seizures in P25 tended to shorten with an increased dose of PTZ (*t* = 1.934, *df* = 13, *p* = 0.0752). The incidence of generalized seizures was 40% higher in P25 animals receiving PTZ in a 100 mg/kg dose compared to 60 mg/kg. However, the difference was not significant (*p* = 0.1490).

Early exposure to Ro 25-6981 had little effect on seizure susceptibility assessed with a PTZ test later in life. The severity of the seizures induced by either dose of PTZ did not differ between the controls and treated animals in P12 (*H* = 0.841, *p* = 0.6566 and *H* = 4.143, *p* = 0.1260, respectively) or P25 animals (*H* = 0.1099, *p* = 0.9466 and *H* = 1.600, *p* = 0.4493, respectively). No effect of early drug exposure on latency to generalized seizures was observed after either dose of PTZ in P12 (*H* = 0.9393, *p* = 0.625, and *H* = 0.5412, *p* = 0.7629, respectively) or P25 (*H* = 0.2573, *p* = 0.3637, and *H* = 0.2481, *p* = 0.2481, respectively). Latencies to minimal seizures in P25 were not affected in animals receiving PTZ in a dose of 60 mg/kg (*H* = 1.581, *p* = 0.4862), but repeated exposure to Ro 25-6981 in a dose of 3 mg/kg significantly shortened latency to minimal seizures after administration of 100 mg/kg PTZ (*H* = 6.382, *p* = 0.0411).

#### 3.2.2. Long-Term Effects on Memory

Effects of early-life exposure to Ro 25-6981 on memory were determined using the passive avoidance test performed in P25 animals (Figure 8). Neither short-term nor long-term memory was affected by repeated drug administration (effect of treatment *F*_(2, 26)_ = 2.480, *p* = 0.1033). Animals of all dose groups remembered the negative experience in the black box in the 1st test and stayed significantly longer in the white box in both the 2nd and 3rd test (effect of interval *F*_(2, 26)_ = 161.3, *p* < 0.0001).

#### 3.2.3. Long-Term Effects on Sensorimotor Development and Behavior in the Open Field

Effects of repeated exposure to Ro 25-6981 on sensorimotor functions and their development was assessed in a battery of motor tests selected according to the developmental stage of the animals (Figure 3). Repeated administration of Ro 25-6981 did not affect latency to righting at any interval after treatment cessation, and all animals returned in their normal upright position in less than 1 s.

Interestingly, in the negative geotaxis test, two-way repeated measure ANOVA revealed a significant effect of interval (*F*_(1, 27)_ = 4.167, *p* = 0.0461) and a treatment × interval interaction (*F*_(2, 27)_ = 3.727, *p* = 0.0305) and showed that repeated administration of 3 mg/kg Ro 25-6981 resulted in a significant shortening of latency to turn at P15 (*q* = 0.0140) (Figure 9).

In the rotarod test, the performance of rats was determined at P25 as the latency to fall from the rod. Performance in this test did not differ between the controls and treated animals (*F*_(2, 27)_ = 1.255, *p* = 0.3011). Performance in the horizontal bar test expressed as latency to stay on the bar was determined repeatedly from P12 till P60. Latency peaked between P18 and P25 and the repeated-measures ANOVA revealed significant effects of interval after treatment cessation (F_(6, 27)_ = 13.62, *p* < 0.0001) and a treatment x interval interaction (*F*_(12, 27)_ = 1.918, *p* = 0.0356). Performance in this test was not affected by repeated drug administration during early development (*F*_(2, 27)_ = 0.5980, *p* = 0.5570).

Possible effects of repeated early life administration of Ro 25-6981 on motor coordination were assessed in the ladder walking test with regular rungs and irregular rungs arrangement. In all treatment groups, latency to cross both regular and irregular ladder improved with age and significantly depended on interval after treatment cessation (*F*_(3, 27)_ = 13.21, *p* < 0.0001 and *F*_(3,27)_ = 29.65, *p* < 0.0001, respectively) and was not affected by early drug exposure (*F*_(2, 27)_ = 0.3096; *p* = 0.7363, and *F*_(2, 27)_ = 1.380, *p* = 0.2698). Independently of treatment, some animals did not cross the ladder, particularly on the 1st day of the test (i.e., at P21). The percentage of P21 animals capable of crossing the regular ladder tended to be higher than the irregular ladder. The difference was not significant and was not affected by repeated administration of Ro 25-6981 (50 vs. 30% in the controls, 70 vs. 50% in the 1 mg/kg group, and 60 vs. 30% in 3 mg/kg group successfully passed the test). The number of errors in both regular and irregular ladder trials decreased with a time interval after treatment cessation (significant effect of time *F*(_1,655, 40,26)_ = 53.37, *p* < 0.0001 and *F*_(1,872, 44,92)_ = 28.99, *p* < 0.0001, respectively) and was not affected by early drug exposure. In all treatment groups, the number of errors was significantly higher in P21 compared to P60 rats in both test modifications.

In the open field test, locomotor activity increased with age in all treatment groups. ANOVA revealed significant effects of time after treatment cessation on the distance to move (*F*_(6, 27)_ = 27.48, *p* < 0.0001) as well as on velocity (*F*_(6, 27)_ = 27.38, *p* < 0.0001). The frequency of visits to the center and time spent in it was considered an index of anxiety. Both parameters were dependent on time interval after treatment cessation; i.e., on the age of the animals during the open-field exposure (frequency of center visit *F*_(6, 27)_ = 13.53, *p* < 0.0001 and time spent in the center *F*_(6, 27)_ = 5.371, *p* = 0.0003). Neither frequency nor time spent in the center was affected by early exposure to Ro 25-6981 (*F*_(2, 27)_ = 0.194, *p* = 0.8247 and *F*_(6, 27)_ = 0.5861, *p* = 0.534, respectively).

## 4. Discussion

In agreement with our expectations, the acute anticonvulsant effects of a selective GluN2B subunit-containing receptor antagonist, Ro 25-6981, were highly age-dependent. In line with our previous study with a less potent antagonist, ifenprodil [17], Ro 25-6981 exhibited more powerful anticonvulsant effects in the PTZ model in infants compared to juvenile rats. In contrast to previous classes of non-specific NMDA blockers, the anticonvulsant action of selective GluN2B antagonists has been studied to a limited extent. In preclinical studies, anticonvulsant effects are routinely tested only in adult animals, and the activity of these drugs in models of reactive seizures in adults is only limited. Pretreatment with ifenprodil was found to decrease the incidence and severity of lindane-induced seizures [28], and Ro 25-6981, in contrast to ifenprodil, was found to suppress MES in adult rats [29]. The available data revealed the superior anti-seizure efficacy of selective GluN2B antagonist in infantile animals compared to older age groups [30]. We previously demonstrated that both ifenprodil and Ro 25-6981 are effective against electrically induced cortical afterdischarges in immature rats. Both drugs were more effective in infantile compared to juvenile rats [16,31]. In infant rats, the acute anti-seizure effects of Ro 25-6981 were also reported in a model of bacterial meningitis [32]. Pronounced anti-seizure activity in this age group of rats is consistent with the change in the amount and distribution of GluN2B subunit during development. In rats, GluN2B-containing NMDA receptors prevail until the second and third week of life, whereas expression of the GluN2A subunit progressively increases and replaces the GluN2B subunit in NMDA receptor assembly with maturation (for a review, see [33]). It has to be mentioned that immature pharmacokinetics can also participate in age-related differences observed in anti-seizure activity [34]. In infantile rats, many systems involved in drug biotransformation or excretion are less active than in adults [35]. This, together with higher permeability of the blood–brain barrier [36], decreased expression, and a lower activity of efflux transporters [37] can result in higher brain levels of Ro 25-6981 in infant rats and consequently in higher anti-seizure activity. In addition to the age-specific profile of anticonvulsant activity, the efficacy of Ro 25-6981 was also highly seizure-specific. In P12 pups, pretreatment with a single dose of Ro 25-6981 resulted in a significant decrease in seizure severity, which was driven mainly through suppression of the tonic phase of generalized seizures. In adult animals, specific suppression of the tonic phase of GTCS by the very high dose of Ro 25-6981 (100 mg/kg) was documented previously in a model of maximal electroshock seizures, which is widely used to screen for anti-seizure drugs active against GTCS [38]. In our study’s dose range, Ro 25-6981 was ineffective against this seizure type in P25 animals. Furthermore, pretreatment with Ro 25-6981 did not affect any measured parameters of minimal seizures except the tendency to shorten the latencies of these seizures. Minimal seizures differ from GTCS by their development course [24], their site of origin, and by pharmacological sensitivity. As suggested by experiments in animals with a brainstem transection, minimal clonic seizures are generated in the basal forebrain [39]. Using the brainstem transection, we previously confirmed this statement also in immature animals. In the same experiments, we demonstrated that generalized tonic–clonic seizures could be induced by PTZ in the transected animals [40]. According to their pharmacological sensitivity, PTZ-induced minimal seizures are with some limitations accepted as a suitable screening model for drugs effective against absence and myoclonic seizures [41].

Early exposure to Ro 25-6981 in doses that significantly decrease seizure severity had no effects on seizure susceptibility within two weeks after treatment cessation, suggesting that anticonvulsant effects do not persist after the end of therapy. At the same time, abrupt treatment termination did not result in the development of seizures or increased seizure susceptibility. These findings suggest that, in contrast to some other anticonvulsant drugs, abrupt interruption of Ro 25-6981 administration does not induce withdrawal in infantile animals [42].

The GluN2B subunit was found to govern various cognitive functions. Numerous studies have shown that inactivating or overexpressing GluN2B, either systemically or in specific forebrain regions, alters various cognitive abilities (for reviews, see [43,44]). Preclinical studies indicate that drugs specifically targeting the GluN2B subunit offer therapeutic benefits without or with only minimal adverse effects (for a review, see [33]). Selective GluN2B antagonists, such as Ro 25-6981, have been found to deteriorate some specific cognitive processes as behavioral flexibility or extinction of memory destabilization in adult animals [45,46]. High perinatal expression of the GluN2B subunit has been taken as evidence for the important role of this subunit in brain development, circuit formation, and maturation of various brain functions [47]. This raises the clinically relevant question of whether pharmacological blockade of the GluN2B subunit during a highly sensitive perinatal period can alter brain functions later in life. Negative functional consequences of early exposure to anti-seizure drugs have been well documented by numerous experimental studies before. Anti-seizure drugs that are routinely used to treat seizures in newborns, such as barbiturates and benzodiazepines, have been found to alter the cognitive, social, and emotional behavior over a long term after treatment cessation [48,49,50]. Similarly, early exposure of developing rats to MK-801, a non-specific antagonist of NMDA, significantly impaired their cognitive abilities in adolescence and adulthood [51,52]. However, data on the possible unwanted consequences of early pharmacological manipulation of the GluN2B subunit for further brain functions and their development are very limited. To determine whether exposure to Ro 25-6981 in the developmental period, which corresponds with the perinatal period in humans, affects brain development and functions, we used a battery of behavioral tests covering all developmental stages until adulthood. The comparability of the developmental stages across species is very complicated. Based on the timing of growth sprout, synaptogenesis, course of myelination, and maturation of neurotransmitter systems, the overall consensus translates the human day of birth to P10–12 rats (for reviews, see [53,54]).

As mentioned above, the detrimental effects on cognitive abilities reported before in animals exposed to non-specific NMDA antagonists early in life represent the major limitation in therapeutic use of these compounds. Early exposure to non-selective NMDA antagonist MK-801 resulted in neurodegeneration [55] and persisting behavioral abnormalities, including cognitive impairment [56,57]. Unlike MK 801, Ro 25-6981 did not exhibit neurotoxic effects in P7 mice and administration of a dose of 10 mg/kg did not induce caspase3 and HSP70 expression [58]. In our study, neither acute nor repeated administration of Ro 25-6981 prevented the juvenile animals to recall the association between the context and the foot shock in the passive avoidance test. As reported previously, three-week-old rats successfully handle this test, and they can still recall the association between the context and the foot shock 24 h after the first exposure [59]. Thus, our study suggests that neither the short-term memory nor the long-term retention of juvenile rats is affected by the administration of Ro 25-6981.

In line with previous studies [58], early exposure to Ro 25-6981 did not affect any evaluated parameter in the open field test, including exploration in the center of the open field box. Furthermore, the increase in frequency or duration of the center visits is taken as a risk behavior, whereas a decrease in these parameters suggests an increased level of anxiety [60]. Taken together, the results of our study suggest that repeated, early life administration of Ro 25-6981 does not substantially alter selected cognitive or emotional abilities later in life. However, this assumption is based only on a limited number of tests. Thus, it is impossible to draw a conclusion concerning the safety of Ro 25-6981 for the development of cognitive abilities and emotional reactivity based only on the results obtained in the present study.

Because behavioral tests typically measure motor responses to sensory information [61], assessing sensorimotor abilities is critical for proper interpretation of animal performance in these tests. In the present study, animals were exposed to an extensive battery of sensorimotor tests covering different developmental stages and motor functions [62,63,64]. Adverse effects of NMDA antagonists in immature animals have been studied only to a limited extent. We previously showed that a single administration of non-specific NMDA antagonists, such as MK-801, CGP 40116, and memantine, compromises sensorimotor performance already in doses effective against acquired seizures in both infantile and juvenile rats [17,27]. In contrast, a single dose of ifenprodil resulted only in mild deficits in sensorimotor performance that quickly faded away [17]. The long-term effects of repeated perinatal administration of these drugs on sensorimotor development were not yet assessed. Our data show that early exposure to Ro 25-6981 does not disturb sensorimotor performance at any time interval after treatment cessation. Even the most demanding motor skills test, such as bar holding [65], was not affected by repeated Ro 25-6981 in any tested dose. In addition, Ro 25-6981 exposure did not alter the coordination of juvenile animals in the rotarod test or the capability of fine-tuning stepping in the ladder rang walking test. These results demonstrate that the selective GluN2B antagonist does not disturb the sensorimotor performance or locomotion development.

## 5. Conclusions

In line with the developmental profile of the GluN2B subunit, anti-seizure activity of a single dose of Ro 25-6981 was detected in infantile but not in juvenile rats. At the same time, repeated administration of Ro 25-6981 in doses effective against PTZ-induced seizures to neonatal rats did not deteriorate the sensorimotor abilities and their postnatal development. Early-life exposure to this drug did not interfere with emotionality or cognitive abilities. Due to the limited number of behavioral tests used in this study, the results have to be confirmed in future studies. Taking together our previous [20] and recent data, we can conclude that GluN2B-preferring or selective antagonist drugs exhibit promising anti-seizure and safety profiles in models of early life seizures.

## Figures and Tables

**Figure 1 pharmaceutics-13-01482-f001:**
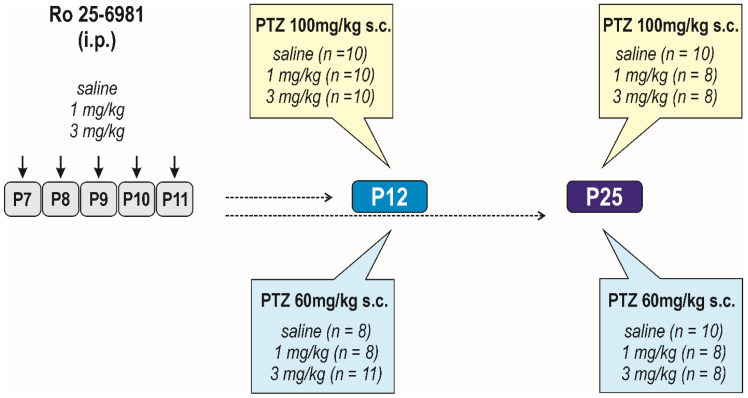
Design/time diagram of seizure susceptibility assessment. Scheme of Ro 25-6981 administration and used doses are shown in the left part of the diagram. Ro 25-6981, dissolved in saline, was administered in two doses, 1 and/or 3 mg/kg i.p. for five consecutive days (P7–P11). Controls received saline in corresponding volume instead of Ro 25-6981 solution. Seizure susceptibility was assessed in two age groups, P12 and P25 animals; i.e., 1 and 14 days after treatment cessation. Seizures were induced with PTZ in either dose 100 and/or 60 mg/kg s.c. only once; i.e., separate groups of animals were used for each age and dose group of animals.

**Figure 2 pharmaceutics-13-01482-f002:**
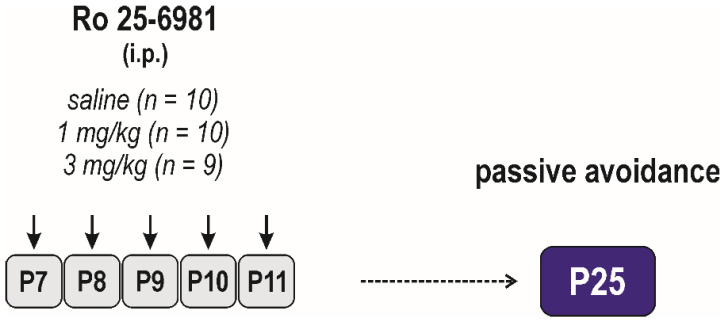
Design/time diagram of the passive avoidance test. Animals were tested at P25, and they were not used for any other test. Other details as in Figure 1.

**Figure 3 pharmaceutics-13-01482-f003:**
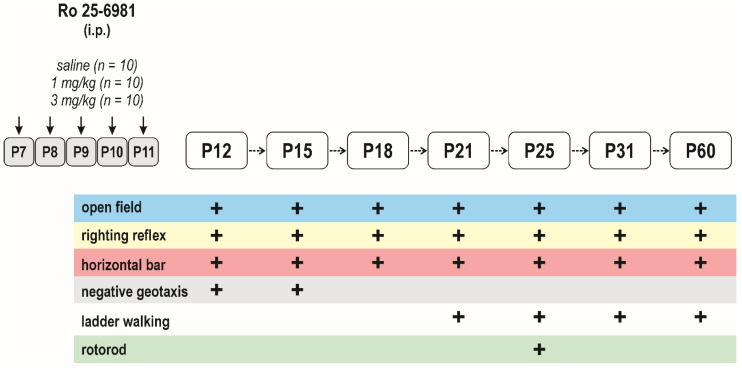
Design/time diagram of the sensorimotor tests: The upper left part of diagram shows the scheme of drug administration, used doses, and number of animals in each dose group. The upper right part of the diagram indicates ages at testing. Animals were tested repeatedly, starting at P12 till P60, using the battery of behavioral tests listed in the lower part of the diagram. Other details as in Figure 1.

**Figure 4 pharmaceutics-13-01482-f004:**
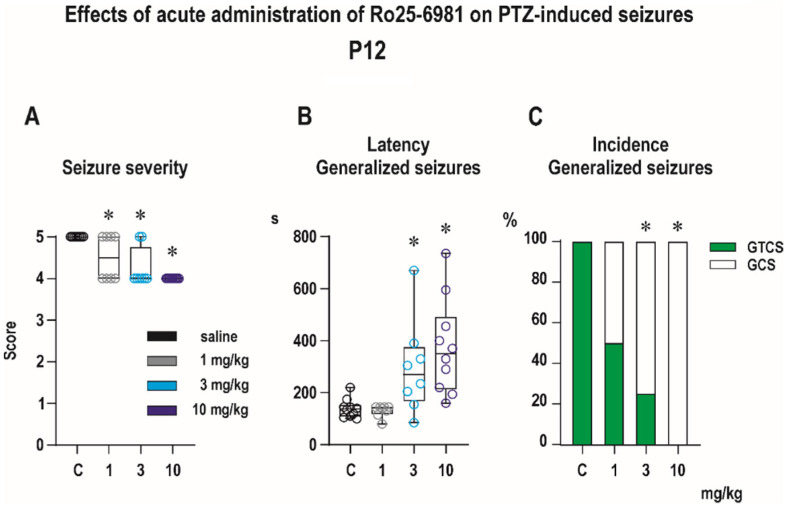
Effects of Ro 25-6981 on pentylenetetrazole (PTZ)-elicited seizures in P12 pups (experimental groups consisted of 8–10 animals). (**A**) Ro 25-6981 in all tested doses significantly decreased the seizure severity expressed as a score (*y* axis). This effect was driven mainly by suppressing the tonic phase (TF) of generalized seizures (GS). (**B**) Administration of Ro 25-6981 in doses 3 and 10 mg/kg i.p. significantly extended latencies (in seconds, *y* axis) to GS onset. (**C**) In the highest dose of 10 mg/kg Ro 25-6981 completely abolished TF, but the total incidence of GS was not affected by pretreatment with Ro 25-6981 in any tested dose. The percentage of animals exhibiting complete generalized tonic–clonic seizures (green) and generalized seizures without tonic phase, i.e., generalized clonic seizures (white) is on the *y* axis. Asterisks denote a significant difference in comparison with the controls. The *x* axis in all three graphs presents the control and three groups with different doses (1, 3, and 10 mg/kg) of Ro 25-6981. Data in graphs (**A**,**B**) are presented as box plots (the sample median and the first and third quartiles) with whiskers (min and max). Circles mark individual values. Asterisk denotes significant difference compared to controls.

**Figure 5 pharmaceutics-13-01482-f005:**
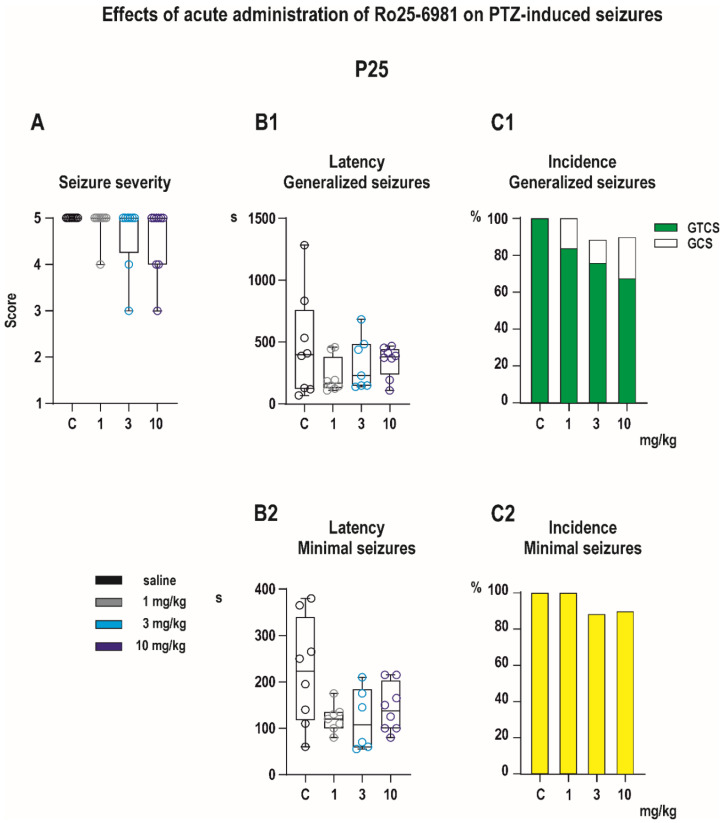
Effects of Ro 25-6981 on PTZ-elicited seizures in P25 pups (experimental groups consisted of 8–10 animals). None of the evaluated parameters of GS, i.e., seizure severity (**A**), latency (**B1**), and incidence of GS or separate TF (**C1**), was affected by administration of Ro 25-6981. Ro 25-6981 in all three doses tended to shorten latencies to minimal seizures, but differences did not reach a level of significance (**B2**). The incidence of this seizure type remained unchanged (**C2**). All details and symbols as in Figure 4.

**Figure 6 pharmaceutics-13-01482-f006:**
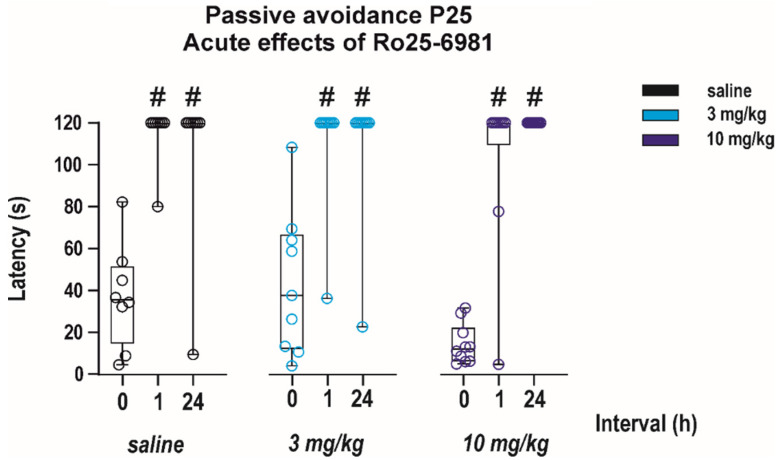
Effects of a single administration of Ro 25-6981 on performance in the passive avoidance test. The drug was administered in two doses (3 or 10 mg/kg i.p.) to P25 animals 30 min before the 1st exposure (interval 0; *x* axis) and the test was repeated two times—1 and 24 h after the 1st exposure. Experimental groups consisted of 8–9 animals. Neither tested dose of Ro 25-6981 affected latencies to step through (in seconds; *y* axis). # denotes the significant difference in latencies to step through between the 1st and repeated tests. All details and symbols as in Figure 4.

**Figure 7 pharmaceutics-13-01482-f007:**
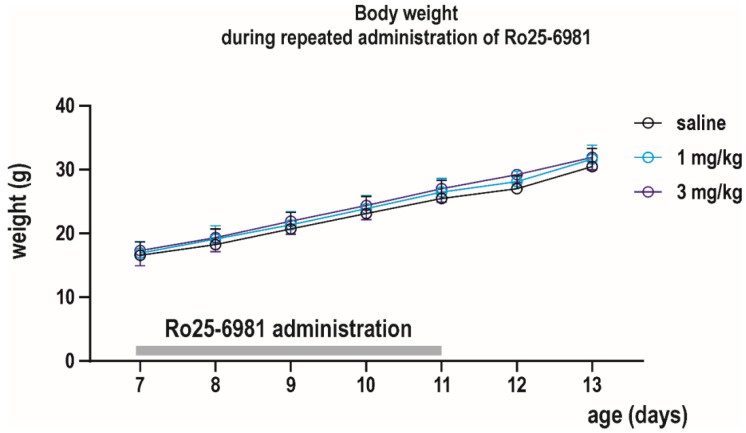
Effects of repeated administration of Ro 25-6981 on body weight. Ro 25-6981 was administered daily at P7–P11 (*x* axis) in two doses of 1 and 3 mg/kg i.p. Body weight was checked daily till P13. Body growth (in grams, *y* axis) was not affected by drug exposure.

**Figure 8 pharmaceutics-13-01482-f008:**
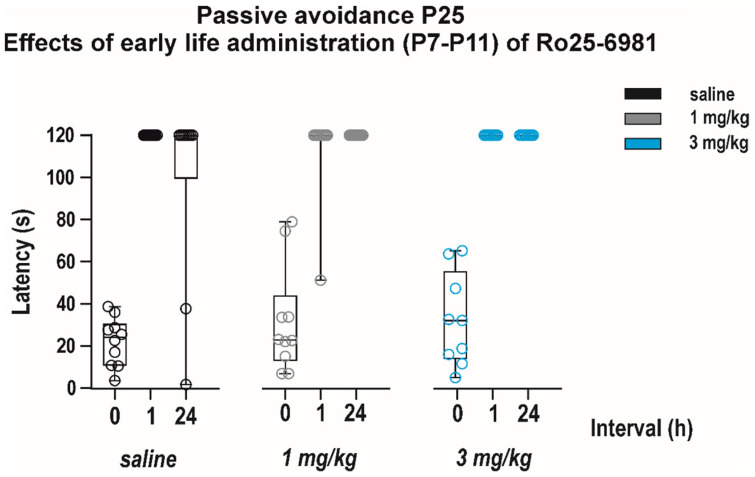
Effects of early life administration of Ro 25-6981 on performance in the passive avoidance test. The drug was administered as described above at P7–P11 in dose 1 and 3 mg/kg i.p. Animals (experimental groups consisted 9–10 animals) were tested at P25; i.e., 14 days after treatment cessation, and the test was repeated three times (for details, see Figure 6). None of the tested doses influenced the performance of animals in this test, suggesting that early life exposure to Ro 25-6981 does not impair cognitive abilities later in maturation. All details and symbols as in Figure 4 and Figure 6.

**Figure 9 pharmaceutics-13-01482-f009:**
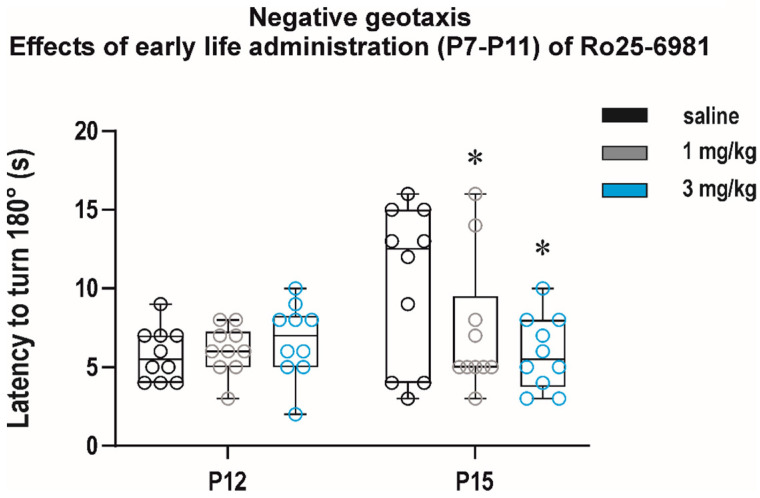
Effects of early life exposure to Ro 25-6981 on performance in the negative geotaxis test. Ro 25-6981 in two doses, 1 and 3 mg/kg, was administered intraperitoneally for five consecutive days, starting at P7 till P11. Each group consisted of 10 animals, and animals were tested after treatment cessation at P12 and P15 (*x* axis). In the 2nd test at P15, animals exposed to Ro 25-6981 in both doses managed to turn 180° in a significantly shorter time than the controls (*y* axis). All details and symbols as in Figure 4.

## Data Availability

The data presented in this study are available on request from the corresponding author.

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
