# Peer review of "The GluN2B-Selective Antagonist Ro 25-6981 Is Effective against PTZ-Induced Seizures and Safe for Further Development in Infantile Rats"

_pharmaceutics, 2021, doi:10.3390/pharmaceutics13091482_

Round 1
Reviewer 1 Report
Mares et al. studied the effect of a GluN2B antagonist (Ro25-6981 at 1, 3 and 10 mg/kg) on the susceptibility of immature rats to pentylenetetrazol (PTZ) induced seizures and on several cognitive and sensorimotor tasks. They concluded that there was an age-specific anticonvulsant effect, in that Ro25-6981 increased the latency and decreased the stage of tonic-clonic seizures on P12, but did not affect seizure susceptibility on P25. There was no long-term effect of repeated injections of Ro25-6981 (P7 to P11) on seizure susceptibility on P12 or P25. Ro25-6981 drug alone did not affect passive avoidance test recall at 1 h or 24 h after a single acute dose, or on P25 following prior repeated injections (P7 to P11). Similarly, righting response, negative geotaxis, bar holding, ladder rung walking, and open field tests were not affected by the repeated injections of Ro25-6981 alone.
This study addressed the age-dependent effect of GluN2B antagonist on seizure susceptibility, and its behavioral and cognitive effects up to 14 days after repeated drug administration. These issues have been fully addressed previously and safety of other GluN2B antagonists is controversial. The study appears to be well done. It used an adequate number of animals (and litters) and an established seizure model, which the researchers have ample experience.
While there appears to be no adverse effects of Ro25-6981 on behavior and cognitive, the anticonvulsant effect of Ro25-6981 also appears to be small. The highest dose of Ro25-6981 used apparently only delayed the onset of generalized seizures or only reduced them to stage 4 (Fig. 4). Whether this may ameliorate the duration of seizure or other consequences of seizures (such as motor and cognitive impairments) is not known. Could the authors comment whether this could be a model specific effect, i.e., using a high dose (100 mg/kg i.p.) of PTZ, or this could this be considered an effective anticonvulsant?
My other comments are minor, and relate to clarifying the presentation, as follows.
- In Introduction (line 31, p.2), the rationale of using Ro25-6981, compared to other antagonists (ifenprodil e.g.,), can be given. In the Discussion, Ro25-6981 was said to be more potent (p.14).
- In Methods, (i) whether the dams were kept in the same animal quarters of the researchers’ institution may be worth mentioning, (ii) does maintaining “the temperature in Plexiglas cages” mean the temperature of the cage floor? (iii). P.3 line 34, seizures are; following sentence (lines 35-37) can append whether occurrence of GTCS declines with age (rare on P15?); (iv) the meaning of tonic in tonic seizures can be briefly described; (v) line2 of p.5, add “receiving Ro 25-6981” after “groups of animals”; (vi) p. 6 line 13, ladder instead of leader.
- In the Results, p.7 line 20, the authors should cite their own data in Fig. 4, perhaps adding that "clonic seizures with preserved righting were not found"; the following sentence of “According to the literature” need not be there.
- p.8, line 13, cite Fig. 5, and continue with lines 14-16 as the same paragraph.
- p.8, line 17, no effect should be “no significant effect”
- p.9, line 9, section 3.1.1 should prepare reader for a different topic, perhaps start with “The effect of Ro 25-6981 alone on passive avoidance test was done in P25 animals.
- p. 9, line 10, to be specific, state 1-hour and 1-day responding instead of short- and long-term.
- Fig. 6 vertical axes should be labeled as “latency (s)”; this reviewer would recommend that all axes be labeled, but this may not be as critical in other figures with describing titles.
- p.11, Section 3.2.1 title should be “seizure susceptibility”, and not “brain excitability”, which is not specific. The same applies to the rest of the text.
- p.11, line 8, Fig x is apparently not found.
- Fig. 9, saline and grey lines are not clearly distinguished, changing grey to another color is suggested; alternatively, regrouping the points with Saline, 3, 10 labels on the horizontal axes may work.
- p.15, meaning of perinatal administration is not clear, and appears to be used only here; was this the P7-11 repeated injection paradigm?
- Discussion, p. 14, line 22 is not a sentence
Author Response
Authors would like to thank for helping them to improve the manuscript and correcting mistakes.
- In fact, high-dose PTZ model, as it is used in our laboratory, covers two seizure types – age-related minimal seizures and generalized tonic clonic seizures that can be induced in all age groups starting at P1. Their pharmacological sensitivity is different and, in addition to complete suppression of GTCS, some anti-seizure drugs (for example primidone, carbamazepine, oxcarbazepine, topiramate or lamotrigine) suppress specifically tonic phase of these seizures. Drugs specifically suppressing tonic phase of GTCS usually also supress tonic extension in MES model. We added brief information into discussion (P14, l 35)
- Rationale of using Ro 25-6981 was added into Introduction (p2, l32-42). Thank you for suggestion.
- (i) Dams with litters were kept in animal room in our department, next door to the room used for behavioural testing. (ii) Temperature is measured inside of the small plastic containers that are used for animal observation. Measuring probe is fixed to the bottom part of container. We added these information into Method part (p 3, paragraph 2). (iii) No, in PTZ model only development of minimal (mS, clonic) seizures is age-dependent, GTCS can be induced in all age groups of animals starting at P1. The sentence was reformulated and we hope that it is clear now. (iv) GTCS compose of three phases (1) wild running, (in animals with immature motor system is replaced by swimming movements, crawling), tonic phase with loss of righting reflexes (tonic flexion of all four paws progressing to tonic extension in some cases) and clonic phase represented by clonic movements and twitches of muscles. (v; vi) This was corrected in text, thank you.
We corrected typos and mistakes and try to clarify the text according to suggestions. Thank you.

Reviewer 2 Report
In this manuscript, the authors reported that a synthetic compound Ro25-6981 can display anti-seizure effects in a rat model. They found that Ro25-6981was effective against PTZ-induced seizure in infantile rats, specifically suppressing the tonic phase of generalized tonic-clonic seizure, but it failed in juveniles. In addition, exposure to Ro25-6981 early in life did not seem to have significant adverse effects on cognitive function and excitability. In general, this study showed a positive conclusion on the potential of Ro25-6981 against seizures, especially in newborn individuals. The research results can be helpful for the development of future strategies for the treatment of neonatal epilepsy. However, most of the conclusions were only obtained by measuring behavioral responses. It would be better if authors can provide some additional histological and even electrophysiological results. In addition, I only have a few comments for the authors to refer.
- In all experiments, the authors used the PTZ only group as the control group. The lack of a sham (vehicle only) group makes it a little difficult to assess what should be the behavior of normal rats.
- In assessing the long-term impact, the authors only analyzed both the passive avoidance test and the geotaxis test. Can these two tests represent the assessment of all brain damage? If there is some additional histological evidence (such as TUNEL analysis of brain slices), it will make the conclusion more solid.
- For all CNS medications, the penetration of BBB is a very important issue. I suggest that authors may consider adding this narrative in the discussion section (e.g. ACS Omega 2019; 4:9925-9931).
Author Response
Many thanks for helping us with typos and mistakes in our manuscript. We corrected them.
According to your suggestion we added brief introduction and rationale for selection of Ro 25-6981 in Introduction (p2, L 32-42). Many thanks for suggesting this.
We apologize for omitting information concerning route of administration of Ro 25 6981. Drug solution was administered intraperitoneally and this information was included into Methods (P3, L27) and added in figures 1-3 that illustrate study design and in figure legends.

Reviewer 3 Report
The present study examined the age-dependent differences in the anticonvulsant activity of GluN2B-selective antagonist Ro25-6981 against pentylenetetrazol (PTZ)-induced convulsive seizures in infantile (P12) and juvenile (P25) rats. Then, this study assessed the safety of Ro25-6981 for the developing brain by administering this drug from P7 till P11 and later performing various behavioral tests for sensorimotor development, cognitive abilities and emotionality. The results demonstrated developmental differences in the anti-seizure activity of Ro25-698 and its safety for the developing brain in infantile rats. There are some points (typos and others) to check as shown in the following:
P2/19
L12: a-amino-3-hydroxy-5-methylisoxazole-4-proprionic acid [AMPA]
L19: NMDA receptors are heterotetramers with two obligatory type 1 subunits (GluN1) and two other subunits. -> As the first sentence in the next paragraph.
*L31: Ro 25-6981 -> Give some introduction
P3/19
*L31: Ro25-6981 administration method?
P5/19
*L7: As in figure x,?
P6/19
*L1: Design/time diagram of seizure susceptibility assessment.? Memory test
P10/19
L7: 3 and/or? 10 mg/kg
*References: Keep one writing format for the title and journal name. Check the following:
P17/19
L21: R15: European Journal of Pharmacology
L30: R20: Anticonvulsant Action of GluN2A-Preferring Antagonist PEAQX in Developing Rats.
L32-33: R21: Effect of 7-Nitroindazole, a Neuronal Nitric Oxide Synthase Inhibitor, on Behavioral 32 and Physiological Parameters. Physiological Research
P18/19:
L21: R32: Animal Models of Seizures and Epilepsy: Past, Present, and Future Role for the Discovery of Antiseizure Drugs.
L25: R34: NMDA Receptor Function During Senescence: Implication on Cognitive Performance.
Author Response
Authors would like to thank for helping them to improve the manuscript and correcting mistakes.
- We apologize for omitting this information in previous version of this manuscript. All controls used for either acute or chronic experiments received solvent i.e. saline in corresponding volume instead Ro 25-6981 solution.
- In our study we analysed effects of early Ro 25-6981 on behaviour of animals using the whole battery of behavioural tests as showed in scheme in Figure 2 and 3. Significant changes in behaviour were observed only in the negative geotaxis test – both tested doses resulted in significant shortening of time necessary to turn compared to control. No other changes were observed in used battery of tests.
- We did not include any experiments on neurotoxicity in our study, because in previously published study no pro-apoptotic effects of Ro 25-6981 in a dose of 10mg/kg was observed in P7 rats (Lina-Ojeda et al, 2013). We added brief paragraph concerning neurotoxicity of Ro 25-6981 into Discussion (p15, 31-33). Many thanks for this recommendation.
